# IgG1 Is the Optimal Subtype for Treating Atherosclerosis by Inducing M2 Macrophage Differentiation, and Is Independent of the FcγRIIA Gene Polymorphism

**DOI:** 10.3390/ijms24065932

**Published:** 2023-03-21

**Authors:** Rui Duan, Yan Liu, Dongmei Tang, Run Lin, Jinrong Huang, Ming Zhao

**Affiliations:** Department of Pathophysiology, Key Lab for Shock and Microcirculation Research of Guangdong, School of Basic Medical Sciences, Southern Medical University, Guangzhou 510515, China

**Keywords:** atherosclerosis, immune therapy, CVI-IgG isoforms, FcγRIIA, cell differentiation

## Abstract

In recent years, it has been established that atherosclerosis is an autoimmune disease. However, little is currently known about the role of FcγRIIA in atherosclerosis. Herein, we sought to investigate the relationship between FcγRIIA genotypes and the effectiveness of different IgG subclasses in treating atherosclerosis. We constructed and produced different subtypes of IgG and Fc-engineered antibodies. In vitro, we observed the effect of different subtypes of IgG and Fc-engineered antibodies on the differentiation of CD14^+^ monocytes from patients or healthy individuals. In vivo, Apoe^−/−^ mice were fed a high-fat diet (HFD) for 20 weeks and administered injections of different CVI-IgG subclasses or Fc-engineered antibodies. Flow cytometry was used to assess the polarization of monocytes and macrophages. Although CVI-IgG4 reduced the release of MCP-1 compared to the other subtypes, IgG4 did not yield an anti-inflammatory effect by induction of human monocyte and macrophage differentiation in vitro. Furthermore, genetic polymorphisms of FcγRIIA were not associated with different CVI-IgG subclasses during the treatment of atherosclerosis. In vivo, CVI-IgG1 decreased Ly6C^high^ monocyte differentiation and promoted M2 macrophage polarization. We also found that the secretion of IL-10 was upregulated in the CVI-IgG1-treated group, whereas V11 and GAALIE exerted no significant effect. These findings highlight that IgG1 is the optimal subtype for treating atherosclerosis, and CVI-IgG1 can induce monocyte/macrophage polarization. Overall, these results have important implications for the development of therapeutic antibodies.

## 1. Introduction

Current evidence suggests that atherosclerosis is an autoimmune disease, with several atherogenic antigens documented, including oxidized low-density lipoprotein (oxLDL), heat shock proteins, and fibrinogen [1,2,3,4]. It has been established that oxLDL is a crucial component in the lesion formation [5]. In addition, innate and adaptive immune responses against oxLDL play important roles during the inflammatory process [6]. It has been shown that macrophages express a family of scavenger receptors that bind and uptake oxLDL particles [7]. Continuous activation of such innate immune responses is believed to be a major cause of atherosclerotic plaque development [8]. An increasing body of evidence suggests that immunization of hypercholesterolemic animals with native or oxLDL significantly reduces the development of atherosclerosis [9,10]. It is also widely thought that therapeutic antibodies against oxLDL can exert an anti-atherosclerotic effect [11].

IgG is the predominant class of antibody produced during humoral immunity and has been studied for its potential as a therapeutic antibody. These antibodies perform their many biological functions through crystallizable fragments (Fc) that can bind to IgG receptors (FcγRs) and complement factors [12]. Four IgG subclasses exist in humans with specific binding profiles to six human FcγRs and their polymorphic variants [13]. It has been established that mice produce four IgG subclasses (IgG1, IgG2a/c, IgG2b, IgG3) and express four FcγRs [14]. Although the binding of IgG to FcγR within humans is fairly well documented, the binding of IgG with antigenic specificity to FcγR is largely unstudied.

Cell-mediated immune responses in atherosclerosis are believed to be Th1-dominated, and the key effector molecule is IgG1 [15]. However, a study revealed that pharmacologic enhancement of Th2 responses reduced atherogenesis in Apoe^−/−^ mice [16]. On the other hand, both Th1 and Th2 cells may promote humoral immune responses to protein antigens and stimulate the production of different Ig isotypes. Current evidence suggests that Th2/IL-4 induces IgG4 in humans but IgG1 in mice [17]. During clinical practice, IgG1 is the most commonly used antibody. Although it has been established that IgG1 exhibits a therapeutic effect against atherosclerosis, the effect of IgG4 remains unclear.

On the flip side, unlike humans, mice have no FcγRIIA receptors. FcγRIIA receptors have been documented in atherosclerotic lesions [18] and associated with several pro-atherosclerotic mechanisms at the molecular level [19,20,21,22]. FcγRIIA reportedly mediates the uptake of oxLDL ICs [20] but not of oxLDL in the absence of anti-oxLDL antibodies [23]. OxLDL IgG ICs have also been demonstrated to promote FcγRIIA receptor-dependent monocyte adhesion, and the release of monocyte chemoattractant protein 1 (MCP-1) [19]. Fcγ receptor deficiency in atherosclerosis-prone Apoe^−/−^ mice has been shown to confer protection against atherosclerosis [22]. Yet the role of the FcγRIIA receptor in human atherosclerosis remains unclear. A previous report showed that subjects with homozygous 131His/His genotype exhibit less advanced peripheral atherosclerosis than those carrying other genotypes [24]. However, no genotype-associated difference was observed in the cardiovascular status of patients with coronary artery disease [25]. Thus, the relationship between immunoglobulin isoforms and their Fc receptor genotype should be clarified before clinical trials for antibody therapy against atherosclerosis.

In a recent study [26], we identified an atherosclerosis-related vascular matrix protein, collagen VI, constructed a recombinant human full-length antibody (IgG1) against it, and found that it was similar to an LDL peptide antibody (IgG1) reported by Jan Nilsson et al. [27]. Significant atherosclerosis plaque regression was observed after treatment with CVI-mAb (IgG1) in Apoe^−/−^ mice, which may be attributed to induction of monocyte and macrophage polarization to exert anti-inflammatory effects. This study investigated the relevance of FcγRIIA receptor polymorphism at position 131His/Arg to atherosclerosis to ensure that this antibody is unaffected by the FcγRIIA gene polymorphism during the treatment of atherosclerosis. Previous studies have demonstrated that broadly protective monoclonal antibodies depend on activating, but not inhibitory, FcγRs for activity [28,29]. Therefore, we constructed Fc-engineered antibodies, namely the V11 mutant and the GAALIE mutant, based on a study by Bournazos S. et al. [30]. V11 promotes antibody binding to FcγRIIB whereas GAALIE enhances the affinity of the antibody to FcγRIIA. Moreover, we analyzed the performance of V11 and GAALIE against lipid-induced inflammation in vivo and in vitro compared to the original wild-type antibody. This was also the focus of this study, which aimed to find the optimal subtype of therapeutic antibody.

## 2. Results

### 2.1. Different IgG Subclasses Binding to Human FcγRs

To assess the binding ability of IgGs to human FcγRs and their polymorphic variants, FcγRI, FcγRIIA (H131 and R131), FcγRIIB, and FcγRIIIA, we used a collection of HEK293T cells transfected with GFP-tagged FcγRs sorted to express comparable levels of each FcγR. Human IgG3 exhibited the strongest binding ability to FcγRIA, followed by IgG2, IgG1, and IgG4. Interestingly, CVI-IgG1 could more strongly bind to FcγRIIB compared to other IgGs. IgG2 showed high affinity for FcγRIIA-H131, FcγRIIA-R131, and FcγRIIIA. IgG4 exhibited a similar affinity for all FcγRs (Figure 1).

### 2.2. FcγRIIA Receptor Allele Frequencies

The genotype and allele frequencies of FcγRIIa-131His/Arg receptor polymorphism for the study population were in Hardy–Weinberg equilibrium. The allele frequencies for His131 and Arg131 were 0·54 and 0·46, respectively. The FcγRIIA-131His/Arg genotype was present in 46% (*n* = 60) of study subjects, and the 131His/His and 131Arg/Arg genotypes in 31% (*n* = 40) and 23% (*n* = 30), respectively (Appendix A), which is consistent with the literature [31].

### 2.3. Humanized IgG3 Increases the Release of oxLDL-induced MCP-1

Since polymorphisms in FcγRIIA could influence the ability of anti-oxLDL antibodies to inhibit oxLDL-induced MCP-1 release in CD14^+^ monocytes (results not shown), we explored whether polymorphisms influenced the effect of CVI-IgG on oxLDL-induced MCP-1 release. The results showed that CVI-IgG3 significantly enhanced oxLDL-induced MCP-1 release in CD14^+^ monocytes since it predominantly binds to FcγRIA. As expected, CVI-IgG4 (Th2 induced) exhibited an bigger inhibitory effect on oxLDL-induced MCP-1 release compared to CVI-IgG3, especially in FcγRIIA-131R/R or FcγRIIA-131H/R individuals (Figure 2B,C).

### 2.4. Different IgG Subclasses Do Not Affect Monocyte Differentiation in Individuals of Different Genotypes

Classical monocytes comprise the majority of circulating monocytes and play a pro-inflammatory role in atherosclerosis. However, in the present study, oxLDL did not significantly promote the differentiation of CD14^+^ into classical monocytes in different genotypes of FcγRIIA (Figure 3). Notably, in the FcγRIIA-131H/H individuals, CVI-IgG3 induced the differentiation of classical monocytes, consistent with the results of MCP-1 (Figure 3A,B), suggesting that IgG3 yields pro-inflammatory effects. However, CVI antibodies of other subtypes yielded no obvious effect. Intermediate monocytes are considered to resemble classical monocytes but possess pro-inflammatory properties [32]. After 7 days of stimulation of human peripheral blood CD14^+^ monocytes with oxLDL, the cells were significantly polarized to intermediate monocytes in the FcγRIIA-131R/R individuals (Figure 3C,D). After the different subtypes of antibodies were treated, the oxLDL-induced increase in intermediate monocytes was not reversed. In FcγRIIA-131H/H or FcγRIIA-131H/R individuals, oxLDL and CVI antibodies had no significant effect on CD14^+^ differentiation into intermediate monocytes (Figure 3A,B,E,F). Non-classical monocytes, also known as alternate-activated monocytes, exert anti-inflammatory effects in vivo. Nonetheless, in the present study, oxLDL and CVI antibodies of different subtypes were ineffective against CD14^+^ monocyte differentiation into non-classical monocytes in individuals with different genotypes (Figure 3).

### 2.5. Different IgG Subclasses Do Not Affect Macrophage Differentiation in Individuals with Different Genotypes

Human cells co-expressing CD68 (an LDL and lectin-binding scavenger protein) and CCR2 (MCP-1 receptor) are considered to be pro-inflammatory M1 macrophages, whereas those co-expressing CX3CR1 (specific receptor for fractalkine) and the scavenger receptor CD206 are anti-inflammatory M2 macrophages [33]. Based on these antigenic characteristics, M1 macrophages are more susceptible to higher MCP-1 concentrations entering vessel wall damage sites and differentiating into foam cells, whereas M2 macrophages exert scavenging activity and suppress inflammation via CD206 [34]. However, in vitro experiments showed increased differentiation of both M1 (CCR2^+^CD68^+^) and M2 (CX3R1^+^CD206^+^) macrophages after CD14^+^ monocytes in the FcγRIIA-131H/H (Figure 4A,B) and FcγRIIA-131H/R (Figure 4E,F) normal individuals were treated with oxLDL for 2 weeks, although the difference was not statistically significant. When co-incubated with oxLDL and different subtypes of antibodies for 2 weeks, all antibodies did not suppress the differentiation of inflammatory M1 macrophages or promote the differentiation of anti-inflammatory M2 macrophages. However, compared to the oxLDL-treated group, the M1/M2 ratio was increased after IgG4 treatment in the FcγRIIA-131H/H individuals. In contrast, in the FcγRIIA-131R/R normal individuals, oxLDL significantly increased differentiation of type M2 macrophages. Nevertheless, oxLDL and the different subtypes of CVI-mAb did not affect the M1/M2 ratio.

In summary, although CVI-IgG4 reduced the release of MCP-1 compared to the other subtypes, IgG4 did not yield a corresponding anti-inflammatory effect by the induction of human monocyte and macrophage differentiation in vitro. These results suggested that antibodies were not associated with genetic polymorphisms of FcγRIIA in the treatment of atherosclerosis.

### 2.6. IgG4 Induces Atherosclerosis Plaque Regression Similar to IgG1 in Apoe^−/−^ Mice

In our above in vitro study, IgG4 could significantly inhibit inflammatory chemokines release (MCP-1), although IgG4 could not induce monocyte and macrophage differentiation. However, the pathogenesis of atherosclerosis is complex and the therapeutic effects of Th2-induced IgG4 in vivo remains further validation. Besides, the therapeutic efficacy of IgG4 was independent of FcγRIIA gene polymorphism. Therefore, we conducted adoptive transfer of IgG1 and CVI-IgG4 to high-fat-diet fed Apoe^−/−^ mice (Figure 5A). We observed that similar to CVI-IgG1, IgG4 decreased the plaque area by 40% (Figure 5B,C).

### 2.7. IgG1 Influences the Polarization of Apoe^−/−^ Mice Ly6C^low^ Monocytes and M2 Macrophages

We next detected the differentiation of monocytes and macrophages in Apoe^−/−^ mice during in vitro studies. The results of monocytes differentiation showed that IgG1 and IgG4 did not influence Ly6C^high^ monocytes. Notably, IgG1 yielded anti-inflammatory effects by promoting the differentiation of Ly6C^low^ monocytes (Figure 5D,E). The ratio of Ly6C^high^/Ly6C^low^ was decreased compared with IgG4 (Figure 5E). We next investigated whether IgG1 or IgG4 could induce the expression of macrophage phenotypic markers. We detected the secretion of IL-1β and IL-10. An upregulation of IL-1β was observed in the HFD and IgG4-treated groups (Figure 5F). The differentiation of M1 macrophages was enhanced in the HFD group (Figure 5H,I). We also observed that IL-10 secretion was upregulated in the IgG1-treated group (Figure 5G). Moreover, IgG1 induced the polarization of M2 macrophages compared to Apoe^−/−^ mice (Figure 5J,K). Consequently, although CVI-IgG4 induced atherosclerotic plaque regression, IgG4 was not as effective as CVI-IgG1 in Apoe^−/−^ mice.

### 2.8. Fc-Engineered Antibodies Binding to Human FcγRs

We constructed two mutant antibodies separately to clarify the role of inhibitory and activating receptors in atherosclerosis. By mutating specific amino acid sites, the mutant antibody V11, G237D/P238D/H268D/P271G/A330R, was constructed, which increased the affinity of the antibody for FcγRIIB (Appendix A). We also constructed the mutant antibody GAALIE, G236A/A330L/I332E (Appendix A), to increase the affinity of the antibody for the activating receptors FcγRIA and FcγRIIA to enhance the antibody’s effector function. To assess the binding affinity of Fc-engineered antibodies for human FcγRs and their polymorphic variants, FcγRI, FcγRIIA (H131 and R131), FcγRIIB, and FcγRIIIA, HEK293T cells were transfected with GFP-tagged FcγRs sorted to express comparable levels of each FcγR. The binding affinity results showed that mutant V11 exhibited a lower affinity for the inhibitory receptor FcγRIIB than wild-type CVI-IgG1 (Figure 6B). However, mutant V11 exhibited higher affinity for the inhibitory receptor FcγRIIB than GAALIE. In contrast, the mutant GAALIE increased the affinity of the antibody for the activating receptors FcγRIA, FcγRIIA-H and FcγRIIA-R compared to wild-type CVI-IgG1 (Figure 6A,C,D). Similarly, all three (wild-type IgG1, V11, and GAALIE) showed a weaker affinity for FcγRIIIA. None of these Fc modifications affected the in vitro target antigen-binding specificity (Appendix A).

### 2.9. Fc-Engineered Antibodies Exhibited a Therapeutic Effect in Atherosclerotic Patients Irrespective of FcγRIIA Polymorphisms

It has been established that CVI-IgG1 (wild-type) could induce the differentiation of monocytes into non-classical monocytes and of macrophages into anti-inflammatory M2 macrophages in Apoe^−/−^ mice. However, the above in vitro experiments showed that CVI-IgG1 did not yield the expected therapeutic effect.

Therefore, based on the Fc segment, we treated PBMCs from atherosclerotic patients with 14 mAb (CVI-IgG1), IgG4, Fc-engineered antibodies V11, and GAALIE of wild-type CVI-IgG1. Notably, these antibodies yielded no significant effect on MCP-1 release from PBMCs in patients with different genotypes (Figure 7A–C). We also examined the differentiation of monocytes in the PBMCs. The results showed that IgG1, IgG4, V11, and GAALIE did not affect the differentiation of classical monocytes, non-classical monocytes, and intermediate monocytes when compared to the CON group (Figure 7D–I). Interestingly, FcγRIIA was expressed on many lymphocytes, including monocytes. However, the mutant GAALIE, which has a refined binding affinity for the activating receptor, did not cause alterations in monocyte differentiation, suggesting that monocyte differentiation is not associated with the FcγRIIA gene polymorphism.

### 2.10. Fc-Engineered Antibodies Do Not Affect Atherosclerosis in Apoe^−/−^ Mice

In our above in vitro study, we established that Fc-engineered antibodies did not affect the release of MCP-1 and the differentiation of monocytes during atherosclerosis. However, the in vivo environment associated with atherosclerosis is complex. Accordingly, whether Fc-engineered antibodies are ineffective should be verified with in vivo experiments. Therefore, we conducted an adoptive transfer of Fc-engineered antibodies (V11 and GAALIE) to high-fat-diet-fed Apoe^−/−^ mice (Figure 8A). In the meantime, we mixed CVI-IgG1 and CVI-IgG4 in a 1:1 ratio to compare its effects with the effects of CVI-IgG1. We found that Fc-engineered antibodies did not decrease the plaque area. On the contrary, CVI-IgG1/IgG4 aggravated atherosclerosis compared to CVI-IgG1 (Figure 8B,C). Moreover, the effect of CVI-IgG1 was consistent with the above results (Figure 5B,C).

We also detected the differentiation of monocytes and macrophages in Apoe^−/−^ mice treated with Fc-engineered antibodies. The results of monocyte differentiation showed that they did not influence Ly6C^high^ monocytes. Notably, IgG1 exerted an anti-inflammatory effect compared to other antibody-treated groups as IgG1 promoted the differentiation of Ly6C^low^ monocytes (*p* = 0.0111, Figure 8D,E). We also observed that V11 increased the ratio of Ly6C^high^/Ly6C^low^ compared with IgG1 (Figure 8E). We also measured the levels of serum IL-1β and IL-10. However, we did not detect the secretion of IL-10. At the same time, we found no difference in secreted IL-1β levels between the HFD group and the other treatment groups (Figure 8F). In contrast, the trend for IL-1β in the V11-treated group was consistent with the ratio of Ly6C^high^/Ly6C^low^, suggesting that V11 could increase the affinity of antibodies for FcγRIIB and hold potential pro-inflammatory functions in atherosclerotic mice.

Consistent with the results of macrophage markers, Fc-engineered antibodies did not affect M1 and M2 differentiation, but CVI-IgG1 increased M2 macrophage differentiation in Apoe^−/−^ mice (Chow group). In summary, CVI-IgG1 could treat atherosclerosis by promoting the differentiation of M2 macrophages more than Fc-engineered antibodies and other subtypes of IgG.

## 3. Discussion

Atherosclerosis is a chronic inflammatory disease with a secondary autoimmune component [35]. Over the years, autoantigen-specific adaptive immune responses have been documented in atherosclerosis patients and animal models [36,37]. These immune responses are involved in atherosclerosis progression and contribute to protecting against atherosclerosis. Despite the success of statins and PCSK9 antibodies, atherosclerosis remains a major risk factor for cardiovascular disease. We previously revealed that the binding of IgG to collagen VIA6 was significantly higher in the sera of atherosclerotic patients than in healthy subjects and established their protective effect through vaccine assays. Subsequently, antibodies with high affinity for CVI were screened by a phage antibody library, and their function in atherosclerosis was investigated. These findings suggested that IgG1 contributes significantly to atherosclerotic plaque regression [27].

Since the discovery more than 30 years ago that immune thrombocytopenia could be improved by injectable immunoglobulin therapy, immunoglobulin (IG) preparations have been widely used in autoimmune and inflammatory diseases [38]. Although the main type of immunoglobulin currently available in preparations is IgG [39,40,41], the presence of different subclasses of human IgG (IgG1, IgG2, IgG3, IgG4) and IG preparations largely reflect the heterogeneity of IgG present in serum. On this basis, we investigated the therapeutic efficacy of different subtypes of mAb used to treat atherosclerosis. Four antibodies with specific binding to CVI were constructed by retaining the same antigen-specific binding site for each subtype and replacing only the Fc segment. IgG3 exhibited significant pro-inflammatory effects in vitro, whereas IgG4 inhibited the release of human MCP-1. However, during animal studies, although plaque regression in mouse aorta similar to that cause by IgG1 was observed with IgG4, IgG4 did not significantly affect monocyte–macrophage differentiation or the release of inflammatory factors in mice.

Furthermore, research over the past decade has highlighted the importance of the Fc segment of IgG for the pro-inflammatory activity of IgG, as it acts as an ‘adapter’ linking adaptive and innate immunity [42]. Both the humoral and cellular arms of the innate immune system can be activated and trigger pro-inflammatory responses following the binding of multiple IgG molecules to their specific target antigens, and it can activate innate immune cells by binding to Fcγ receptors. However, the affinity and specificity of FcγRs for different IgG subclasses vary considerably, reflecting the different biological effects of each subclass in triggering different cell types [43]. It is widely acknowledged that the FcγR family consists of several activating receptors and one inhibitory receptor in mice and humans; these receptors are expressed in most innate immune cells, including neutrophils, mast cells, monocytes and macrophages. However, most FcγRs have a low affinity for monomeric IgG and can only be activated in response to multimeric IgG molecules, such as those in immune complexes. In addition, FcγRI is a high-affinity receptor for IgG and recognizes monomeric forms of specific IgG subclasses (IgG1, IgG3 and IgG4 in humans and IgG2a in mice). Depending on the mouse model used, activation of FcγR alone or in combination promotes the immunomodulatory activity of IgG. In this respect, the activity of mouse IgG1 antibodies is dependent on FcγRIII, while FcγRIV or the combination of FcγRIV with FcγRI or FcγRIII is essential for the function of the IgG2a and IgG2b subclasses [42,44,45,46]. Based on these results, studies on human patients receiving tumor antigen-specific IgG antibodies have shown that treatment outcomes could be improved if patients carried engineered antibodies against alleles of human FcγR with a higher affinity for the therapeutic antibody [47,48,49]. Accordingly, we examined the affinity of CVI-mAb for FcγRs and found that the affinity between our constructed CVI-IgG1 and FcγRIIB was the strongest, followed by IgG4, IgG3 and IgG2. In contrast with the literature [50,51], our results showed that the affinity of FcγRIIA for different subtypes of CVI-mAb was in the following order: IgG2 > IgG1 > IgG3 > IgG4. Based on the results of in vitro experiments [52] and the reported literature, the role of FcγRIIB (the only inhibitory receptor) in the treatment of atherosclerosis is far more important than the activating receptor. We also constructed two Fc-engineered antibodies, V11 (exhibiting high affinity for the inhibitory receptor) and GAALIE (exhibiting high affinity for the activating receptor), based on CVI-IgG1 and found that the therapeutic effects of V11 and GAALIE were significantly less than wild-type CVI-IgG1. In summary, our constructed CVI-IgG1 exhibited the best performance among the four isoforms, exhibiting a high affinity for FcγRIIB, reducing the serum levels of IL-1β in mice, inhibiting the differentiation of Ly6Chigh monocytes and enhancing the polarization of M2 macrophages and the levels of serum IL-10 in mice.

In conclusion, in addition to FcγR polymorphisms and IgG subclass responses, many variables affect IgG effector function and clinical outcome. Strong effector functions and high binding affinities, such as FcγR profiles, FcγRIIC-ORF, large numbers of FcγRIII and FcγRIIC gene copies, IgG3-G3m15/16, and non-lamellar glycosylated IgG variants [53], may be associated with “good” outcomes in the prevention of infectious diseases or antibody-mediated therapy. However, these may also become “worse” in autoimmune-mediated diseases. Accordingly, these factors should be considered when studying biological responses from a diagnostic and therapeutic perspective.

## 4. Materials and Methods

### 4.1. Construction and Production of IgG and Fc Variant Antibodies

The variable region for the humanized anti-CVI antibody was generated as previously described [27]. The heavy chain variable region (VH) and light chain variable region (VL) genes were cloned into the modified mammalian expression vectors pcDNA3-IgG1 and pcDNA3-κ, respectively, using the One Step Cloning Kit (Vazyme, CN). The vectors for the pcDNA-VHCγ plasmid and the IgG2, IgG3, and IgG4 heavy chains were cleaved with restriction endonucleases Sal I and BamH I (NEB, USA). The Fc fragment of the original IgG1 was replaced by that of IgG2, IgG3, and IgG4 (Appendix A). The site-directed Fc-engineered antibodies were introduced into the heavy chain using a Mut Express IIFast Mutagenesis Kit V2 (Vazyme, China) according to the manufacturer’s instructions. The DNA sequence was confirmed using SnapGene software (Appendix A).

### 4.2. Recombinant Human Monoclonal Different IgG Subclasses Expression

Vectors pcDNA3-IgGs and pcDNA3-κ were co-transfected into 293F cells using the Expi293 Expression System (Invitrogen, Waltham, MA, USA). The full-length human IgG κ chain was purified from a serum-free medium on an rProtein G 4FF chromatography column (YENSEN Biotech Co., Ltd., Shanghai, China) using an HPLC instrument (AKTA, GE Healthcare). Purified antibodies were concentrated through 30KD ultra-filtration tubes.

### 4.3. Characterization of CVI mAb

The antigenic specificity of recombinant anti-CVI mAbs and mutants was assessed by conducting in vitro assays using ELISA (Appendix A). The purity of recombinant antibodies was assessed by SDS-PAGE and Coomassie brilliant blue R-250 staining (Appendix A). All antibody preparations were >90% pure. The endotoxin detection Limulus kit (Bioendo, Xiamen, China) was used to analyze the endotoxin content of CVI-mAb. An endotoxin content of less than 0.05 EU/mL was used in this study (Appendix A).

### 4.4. Determination of Fc Receptor Binding Affinities

HEK293T cells in DMEM with 10% FBS were transfected with the plasmid of human FcγRs (FcγRIA, FcγRIIB, FcγRIIA-H, FcγRIIA-R, and FcγRIIIA) and lipofectamine 3000 (Invitrogen, USA) according to the manufacturer’s instructions. Three days later, cells were collected and treated with CVI-mAb or mutants. IgG complexes were formed as F(ab′)2-aggregated IgG complexes, performed by incubating 10 μg/mL IgG with 5 μg/mL phycoerythrin (PE)-labeled anti-human-Fab-specific goat F(ab)2 fragments in PBS 0.05% BSA 2 μM EDTA, pH7.4 for 30 min at room temperature. A total of 2 × 10^5^ HEK293T cells expressing human GFP-tagged FcγRs were incubated with either of these IgG complexes for 1 h at room temperature. Binding was evaluated using a FACSCanto II flow cytometer (BD Biosciences, San Jose, CA, USA), and the mean fluorescence intensity (MFI) was plotted as a function of antibody concentration using GraphPad Prism software (GraphPad Software, San Diego, CA, USA) from which half-maximal binding (EC50) values were determined by sigmoidal dose-response modeling.

### 4.5. Preparation of oxLDL

Healthy human plasma was obtained from Nanfang Hospital. The LDL (d = 1.020–1.063 g/mL) was isolated by sequential ultracentrifugation and incubated with 5 μmol/L CuSO4 at 37 °C for 24 h, as described previously, to produce oxLDL [54]. OxLDL was sterilized, stored at 4 °C in the dark, and used within 2 weeks. The protein concentration was determined by the Bradford method [55].

### 4.6. Preparation of CD14^+^ Human Monocytes

Normal white blood cells were obtained from the Guangzhou Central Blood center. Monocytes were prepared from PBMCs, as described previously [56]. Briefly, anti-coagulant venous blood from healthy volunteers was layered over the Ficoll-Paque density gradient reagent (Pharmacia, Germany). Next, the human monocytes were purified with anti-CD14 microbeads (Miltenyi Biotec, Germany). The study was conducted according to the principles of the Helsinki Declaration. Written informed consent was obtained from all donors prior to treatment.

### 4.7. Genotyping of FcγRIIA 131His/Arg

Genomic DNA extraction was performed using a FastPure Blood/Cell/Tissue/Bacteria DNA Isolation Mini Kit (Vazyme, Shanghai, China) according to the manufacturer’s instructions. FcγRIIA genotype at position 131His/Arg was determined from genomic DNA by polymerase chain reaction (PCR) [57] using a ClonExpress MultiS One Step Cloning Kit (Vazyme, China) according to the manufacturer’s instructions. PCR amplification was performed using the sense and antisense primers, 5′-GCATCTTCATTTCTGTCTGCCA-3′ and 5′-CAGTGCCCAATTTTGCTGCT-3′ respectively. All DNA sequences were confirmed by SnapGene software (Appendix A).

### 4.8. MCP-1 ELISA

CD14^+^ monocytes were treated with oxLDL (100 μg/mL) with or without pretreatment with CVI-mAb (100 μg/mL) for 48 h. Human MCP-1 ELISA Kits (BD, USA) were used to assay the cell culture medium. The cells were cultured under standard tissue culture conditions.

### 4.9. Flow Cytometry

Single-cell suspensions were first incubated with FcγR Blocking Reagent (Biolegend) in staining buffer (2% FBS, 0.1% Na-azido, in PBS) for 10 min at 4 °C to block binding to Fc receptors. Then the cells were stained with the corresponding antibodies in a staining buffer for 30 min at 4 °C according to the manufacturer’s instructions. After washing twice in staining buffer, the cells were subjected to flow cytometry analysis (BD Fortessa II, FACSVerse or Canto II), and the data were analyzed using FlowJo v10.4. To analyze the differentiation of monocytes and macrophages in human PBMCs, sorted CD14^+^ monocytes were cultured with oxLDL with or without CVI-mAb. Seven days later, cells were collected and stained with anti-CD14 antibody and anti-CD16 antibody for testing of monocytes. Two weeks later, cells were collected and stained with anti-CD206 antibody, anti-CX3CR1 antibody, anti-CCR2, and anti-CD68 antibody for testing macrophages. Murine monocytes from the peripheral blood were stained with anti-mouse CD11b-PE, CD115-PE-Cyanine7, and Ly6C-PE-Cy5.5 to evaluate the Ly6Chigh and Ly6Clow monocyte populations. Murine macrophages from the peritoneum were stained with anti-mouse F4/80-FITC, CD11c-APC-Cy7, and CD206-AlexaFluor 647 (or CD206-PE) to detect macrophage polarization and finally subjected to flow cytometry (BD Biosciences).

### 4.10. Animal Model and CVI-mAb Treatment

Four-week-old male Apoe^−/−^ mice with C57BL/6 background were fed a high-fat diet (21% fat, 0.15% cholesterol) for 20 weeks [58]. Their diet was changed to normal chow one week before the first immunization. At 25 weeks of age, mice in the CVI-mAb (IgG1, IgG4, V11, and GAALIE) and HFD groups received intraperitoneal injections of CVI-mAb (1 mg/0.5 mL) and PBS (0.5 mL), respectively [59]. After two additional injections at 26 and 27 weeks of age, mice were sacrificed at 29 weeks for the following analyses.

### 4.11. Staining of the Descending Aorta

All mice were humanely killed at the age of 25 weeks. After whole-body perfusion with PBS followed by HistoChoice (Sigma, St. Louis, MO, USA), the aortic arch was dissected and stuck to slides for Oil red O staining and atherosclerotic plaque quantification by computer-aided microscopy using Image Pro software (Zeiss, Germany) as described previously [26].

### 4.12. Serum Analyses

Mouse serum samples were collected and then centrifuged for 10 min at 3000 rpm at 26 °C to separate the mouse serum from the blood cells. The samples were used to analyze mouse serum LDL-C, CHO, HDL-C, and triglyceride (TG) levels which were measured using detection assay kits (Nanjing Jiancheng, China), according to the manufacturer’s protocol. Mouse serum IL-1β and IL-10 levels were detected by detection assay kits (BD, USA) according to the manufacturer’s protocol.

### 4.13. Statistical Analysis

The differences in genotype and allele frequency in the study population were tested using the chi-square test (χ^2^-test) [31]. Homogeneity of variance was tested using Bartlett’s test. All the tests performed were two-sided.

The differences in the means of the MCP-1 and flow cytometry results between FcγRIIa-131His/Arg genotypes in the whole study population and in the subgroups were tested by one-way ANOVA followed by the Brown–Forsythe and Welch multiple comparisons test. The difference in the means of all results in mice was tested using one-way ANOVA followed by Tukey’s multiple comparisons test.

GraphPad Prism 9 was used to generate charts and perform statistical analyses. For all bar graphs, data were represented as mean ± SD. A *p*-value less than 0.05 was statistically significant.

## Figures and Tables

**Figure 1 ijms-24-05932-f001:**
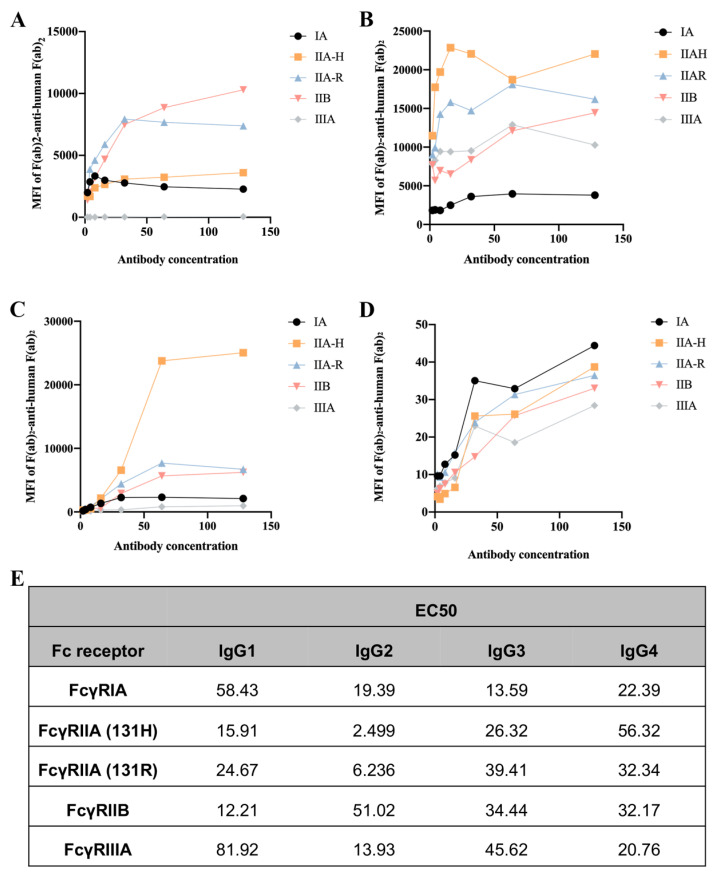
Affinity between the CVI recombinant antibodies of different subtypes and FcγRs. (**A**) Affinity of IgG1 for FcγRs. (**B**) Affinity of IgG2 for FcγRs. (**C**) Affinity of IgG3 for FcγRs. (**D**) Affinity of IgG4 for FcγRs. (**E**) EC50 results of FcγR binding to the CVI recombinant antibodies of different subtypes. *n* ≥ 3 independent experiments.

**Figure 2 ijms-24-05932-f002:**
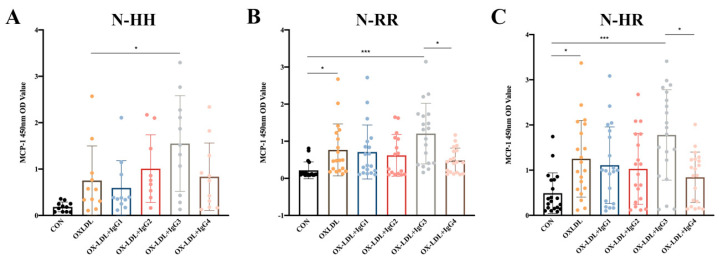
Effects of oxLDL and different isoforms of antibodies on MCP-1 release from human peripheral blood CD14^+^ monocytes. (**A**) Release of MCP-1 in FcγRIIA-131H/H normal individuals (*n* = 11); (**B**) Release of MCP-1 in FcγRIIA-131R/R normal individuals (*n* = 19); (**C**) Release of MCP-1 in FcγRIIA-131H/R normal individuals (*n* = 20). Each ‘n’ number represents cells from PBMCs of a different patient. Data are expressed as mean ± SD. One-way ANOVA followed by Tukey’s multiple comparisons test for (**A**–**C**), * *p* < 0.05, *** *p* < 0.001.

**Figure 3 ijms-24-05932-f003:**
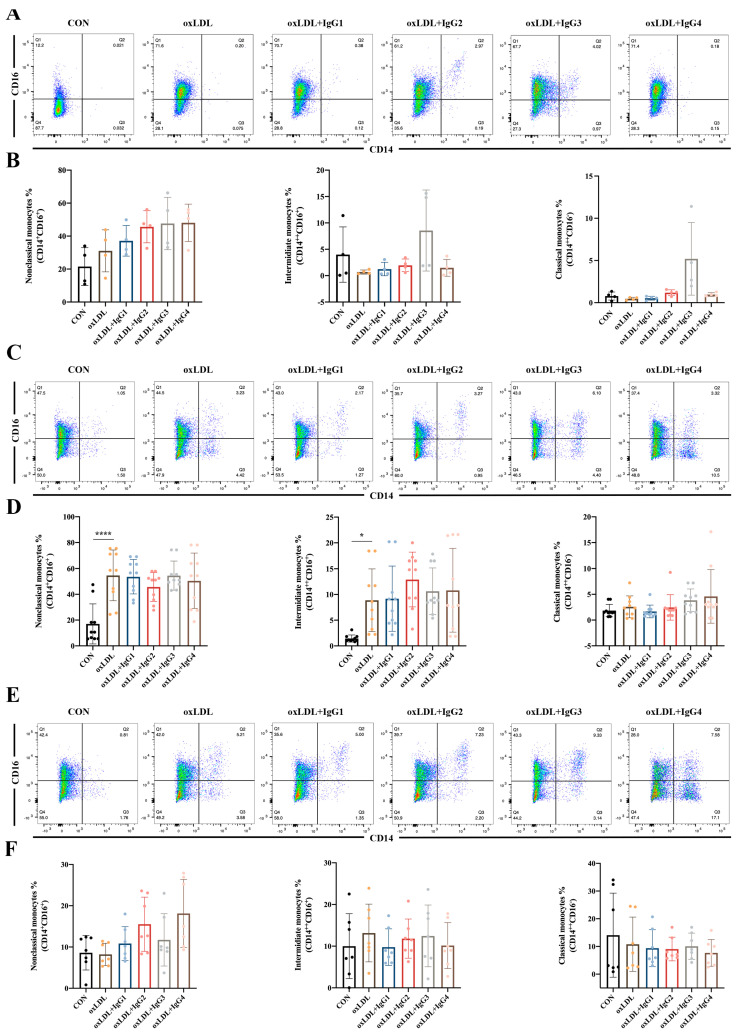
Effect of oxLDL and different subtypes of CVI antibodies on monocyte differentiation in individuals with different genotypes. (**A**,**B**) Flow cytometry analysis of monocyte differentiation in FcγRIIA-131H/H normal individuals (*n* = 4); (**C**,**D**) Flow cytometry analysis of monocyte differentiation in FcγRIIA-131R/R normal individuals (*n* = 10). (**E**,**F**) Flow cytometry analysis of monocyte differentiation in FcγRIIA-131H/R normal individuals (*n* = 7). Each ‘n’ number represents cells from PBMCs of a different patient. Data are expressed as mean ± SD. One-way ANOVA followed by Tukey’s multiple comparisons test for (**B**,**D**,**F**), * *p* < 0.05, **** *p* < 0.0001.

**Figure 4 ijms-24-05932-f004:**
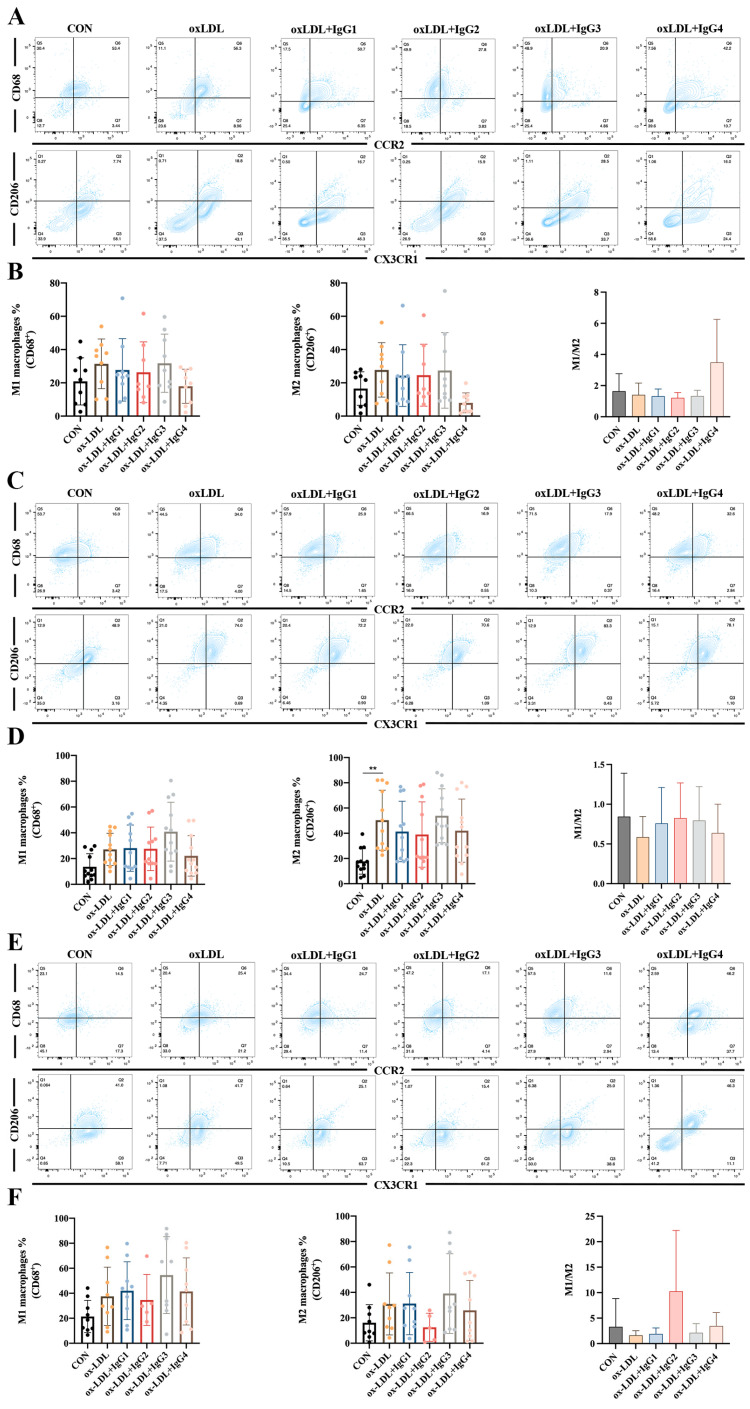
Effect of oxLDL and different subtypes of CVI antibodies on macrophage differentiation in individuals with different genotypes. (**A**,**B**) Flow cytometry analysis of the percentage of M1 and M2 differentiation and the ratio of M1 and M2 in FcγRIIA-131H/H normal individuals (*n* = 9). (**C**,**D**) Flow cytometry analysis of the percentage of M1 and M2 differentiation and the ratio of M1 and M2 in the FcγRIIA-131R/R normal individuals (*n* = 12). (**E**,**F**) Flow cytometry analysis of the percentage of M1 and M2 differentiation and the ratio of M1 and M2 in the FcγRIIA-131H/R normal individuals (*n* = 9). The percentage of M1 and M2 differentiation and the ratio of M1 and M2 in the FcγRIIA-131H/R normal individuals. Each ‘n’ number represents cells from PBMCs of a different patient. Data are expressed as mean ± SD. One-way ANOVA followed by Tukey’s multiple comparisons test for (**B**,**D**,**F**). ** *p* < 0.01.

**Figure 5 ijms-24-05932-f005:**
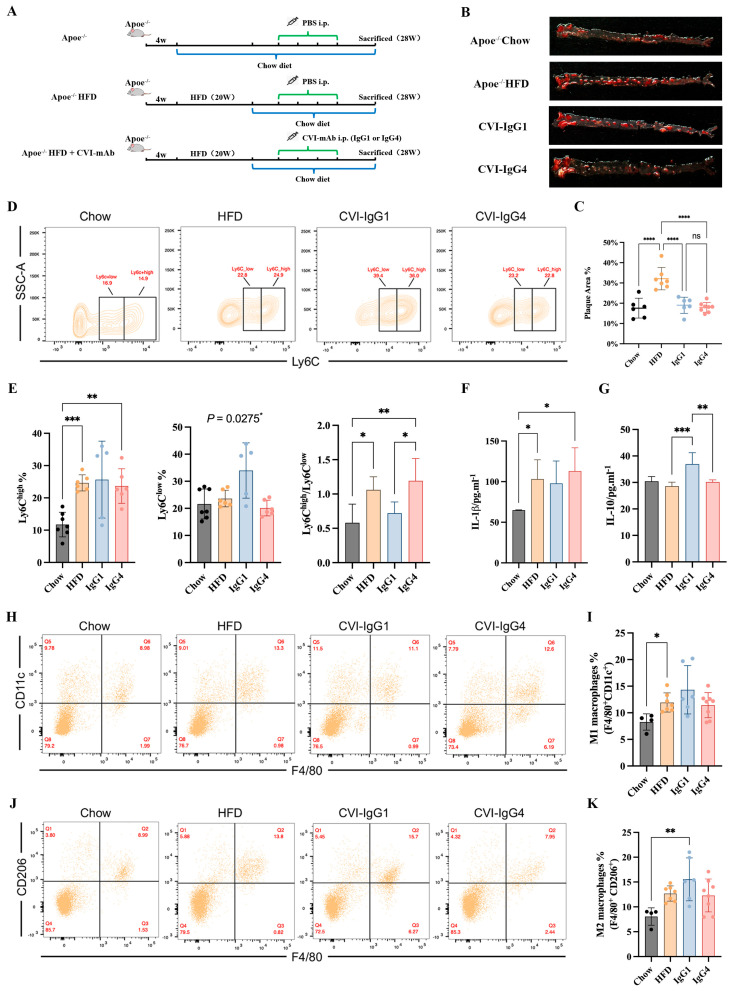
Anti-CVI-mAb adoptive transfer induced HFD-induced atherosclerotic plaque regression in Apoe^−/−^ mice. (**A**) Experimental scheme for high-fat diet-fed mouse and antibody therapy strategy. (**B**) Aortic Oil red O staining of different groups. (**C**) Percentage of plaque area of the entire artery (Chow: *n* = 6, HFD: *n* = 7, IgG1: *n* = 6, IgG4: *n* = 8). (**D**) Flow cytometry analyzed monocytes in different groups. (**E**) Percentage of monocytes (Ly6C^high^ and Ly6C^low^) in different groups and the Ly6C^high^/Ly6C^low^ monocyte ratio in different groups (Chow: *n* = 7, HFD: *n* = 6, IgG1: *n* = 5, IgG4: *n* = 6). (**F,G**) Serum IL-1β and IL-10 levels were measured by ELISA (Chow: *n* = 6, HFD: *n* = 7, IgG1: *n* = 5, IgG4: *n* = 5). (**H**,**J**) Flow cytometry analyzed peritoneal M1 and M2 macrophages in different groups. (**I**,**K**) Percentage of M1 (F4/80^+^CD11c^+^) and M2 (F4/80^+^CD206^+^) macrophages in different groups (Chow: *n* = 4, HFD: *n* = 7, IgG1: *n* = 6, IgG4: *n* = 8). Data are expressed as mean ± SD. One-way ANOVA followed by Tukey’s multiple comparisons test, * *p* < 0.05, ** *p* < 0.01, *** *p* < 0.001, **** *p* < 0.0001.

**Figure 6 ijms-24-05932-f006:**
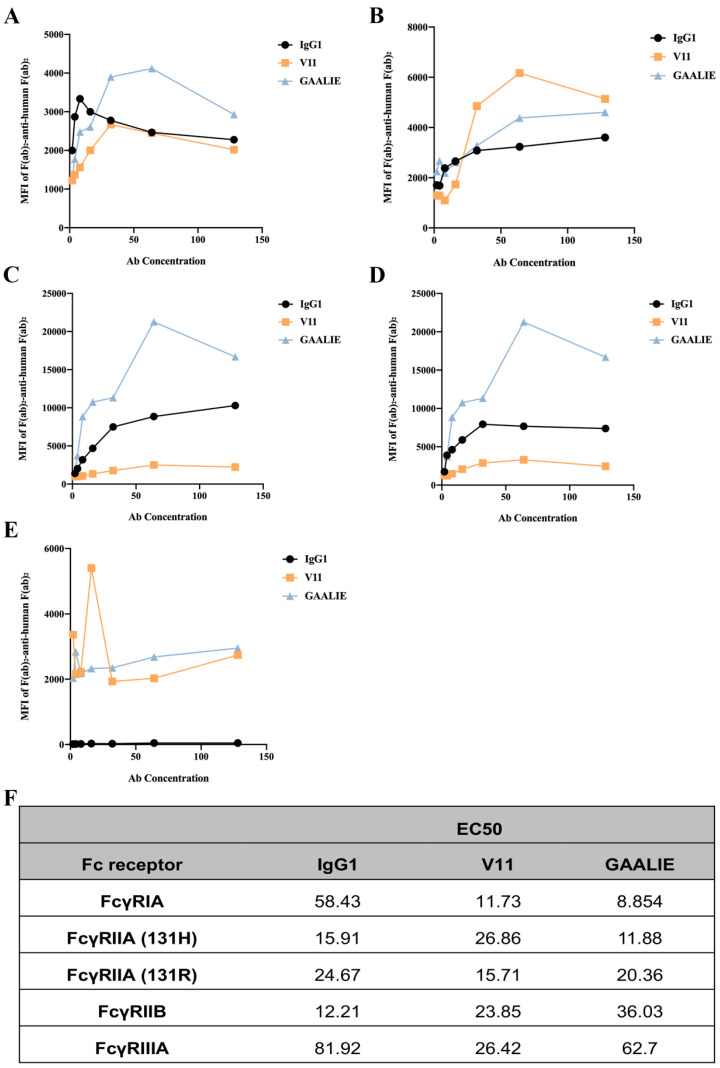
Affinity between Fc-engineered antibodies and FcγRs. (**A**) Affinity of FcγRIA for Fc-engineered antibodies. (**B**) Affinity of FcγRIIB for Fc-engineered antibodies. (**C**) Affinity of FcγRIIA-H for Fc-engineered antibodies. (**D**) Affinity of FcγRIIA-R for Fc-engineered antibodies. (**E**) Affinity of FcγRIIA for Fc-engineered antibodies. (**F**) The EC50 results of FcγR binding to the Fc-engineered antibodies. *n* ≥ 3 independent experiments.

**Figure 7 ijms-24-05932-f007:**
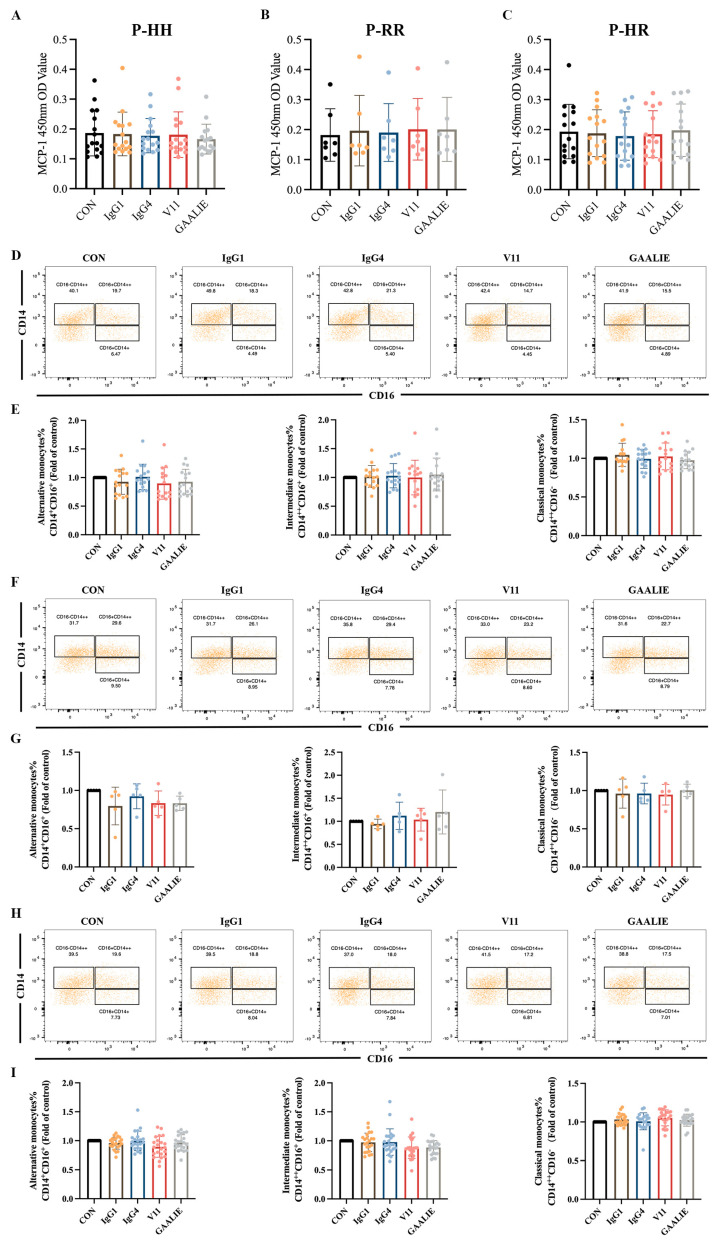
Effects of different isoforms of antibodies and mutants on PBMCs of patients. (**A**) Release of MCP-1 in FcγRIIA-131H/H patients (*n* = 16). (**B**) Release of MCP-1 in FcγRIIA-131R/R patients (*n* = 7). (**C**) Release of MCP-1 in FcγRIIA-131H/R patients (*n* = 15). (**D**,**E**) Flow cytometry analysis of monocyte differentiation in FcγRIIA-131H/H patients (*n* = 15). (**F**,**G**) Flow cytometry analysis of monocyte differentiation in FcγRIIA-131R/R patients (*n* = 5). (**H**,**I**) Flow cytometry analysis of monocyte differentiation in FcγRIIA-131H/R patients (*n* = 22). Data are expressed as mean ± SD. One-way ANOVA followed by Tukey’s multiple comparisons test.

**Figure 8 ijms-24-05932-f008:**
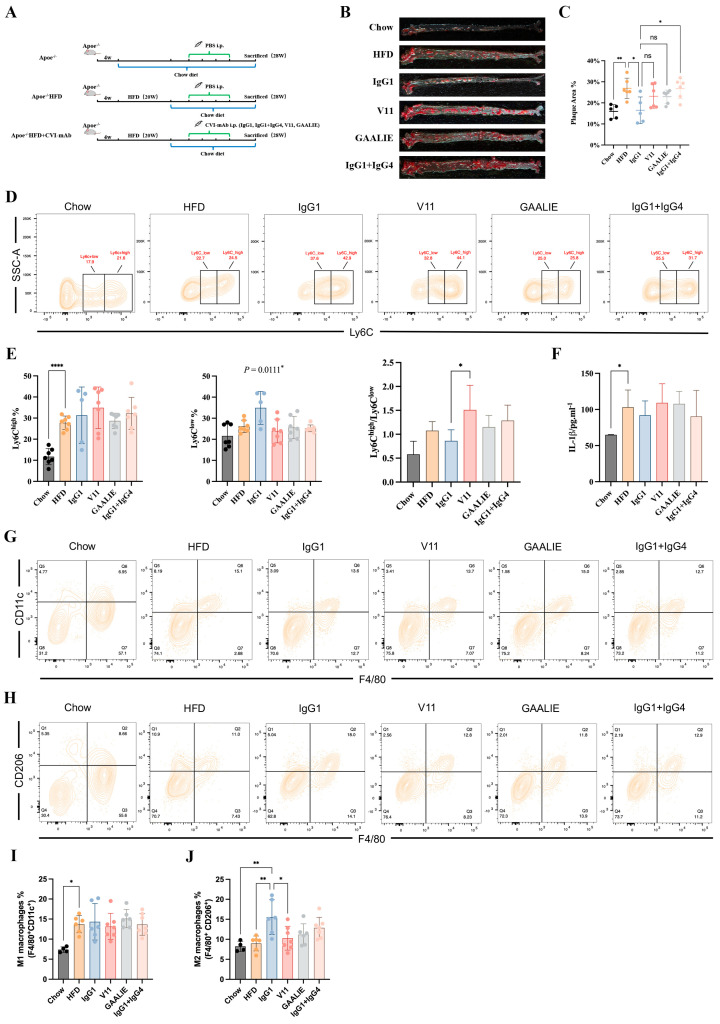
CVI-IgG1 reduced atherosclerotic plaque area in Apoe^−/−^ mice. (**A**) High-fat diet-induced mouse model experimental protocol and antibody therapy strategy. (**B**) Aortic Oil red O staining in different groups. (**C**) Percentage of plaque area of the entire artery (Chow: *n* = 5, HFD: *n* = 6, IgG1: *n* = 5, V11: *n* = 6, GAALIE: *n* = 6, IgG1/IgG4:*n* = 6). (**D**) Flow cytometry analysis of different groups of monocyte (Chow: *n* = 7, HFD: *n* = 7, IgG1: *n* = 5, V11: *n* = 7, GAALIE: *n* = 7, IgG1/IgG4: *n* = 7). (**E**) Percentage of monocytes (Ly6C^high^ and Ly6C^low^) in different groups and ratio of Ly6C^high^/Ly6C^low^ in different groups. (**F**) Measurement of serum IL-1β levels by ELISA (Chow: *n* = 6, HFD: *n* = 7, IgG1: *n* = 5, V11: *n* = 7, GAALIE: *n* = 7, IgG1/IgG4:*n* = 7). (**G**,**H**) Flow cytometry analysis of different groups of intraperitoneal M1 and M2 macrophages (Chow: *n* = 4, HFD: *n* = 6, IgG1: *n* = 6, V11: *n* = 7, GAALIE: *n* = 7, IgG1/IgG4:*n* = 7). (**I**,**J**) Percentage of M1 (F4/80^+^CD11c^+^) and M2 (F4/80^+^CD206^+^) macrophages in different groups. Data are expressed as mean ± SD. One-way ANOVA followed by Tukey’s multiple comparisons test, * *p* < 0.05, ** *p* < 0.01, **** *p* < 0.0001.

## Data Availability

Data are available upon reasonable request. The data that support the findings of this study are available from the corresponding author upon reasonable request.

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
