# Peer review of "IgG1 Is the Optimal Subtype for Treating Atherosclerosis by Inducing M2 Macrophage Differentiation, and Is Independent of the FcγRIIA Gene Polymorphism"

_ijms, 2023, doi:10.3390/ijms24065932_

Round 1
Reviewer 1 Report
The paper is very interesting and well written. The structure is adequate, the endpoints are well defined. The methodology is adequate and coernet with the endpoints of the study. Results are well described. The discussion is coerent with the results and endpoints. I suggest to discuss the role of VEGF and free radicals in the development of atherosclerosis (see and add as references papers by Murdaca et al concerning these topics) and the potential efficacy of biologics as TNF alpha inhibitors on accelerated atherosclerosis (see and add as references papers by Murdaca et al).
Reviewer 2 Report
Dear Authors,
presented work is interesting
Lin 83-85 should be rewritten to be more clear
line 102 titel should be correctted
line 409 healthy human serum meaning
line 473 statsitic should be bettre explained- on what type of data which statistic i suitable
line 468 m LDL- 468 C, CHO, HDL-C and triglyceride (TG) why these parameters
in results there is part about patient teratement however in methodology there is no part how patients were selected. ist there aethic approval (for mice should be also clear stated)
Best of luck
